# Effects of Intermediate Frequency (150 kHz) Electromagnetic Radiation on the Vital Organs of Female Sprague Dawley Rats

**DOI:** 10.3390/biology12020310

**Published:** 2023-02-14

**Authors:** Venkatesan Sundaram, Stephanie Mohammed, Brian N. Cockburn, M. R. Srinivasan, Chalapathi R. Adidam Venkata, Jenelle Johnson, Lester Gilkes, Kegan Romelle Jones, Nikolay Zyuzikov

**Affiliations:** 1Department of Basic Veterinary Sciences, School of Veterinary Medicine, Faculty of Medical Sciences, The University of the West Indies, St. Augustine 999183, Trinidad and Tobago; venkatesan.sundaram@sta.uwi.edu (V.S.); lester.gilkes@sta.uwi.edu (L.G.); 2Department of Physics, Faculty of Science and Technology, The University of the West Indies, St. Augustine 999183, Trinidad and Tobago; stephanie.mohammed@sta.uwi.edu (S.M.); nikolay.zyuzikov@sta.uwi.edu (N.Z.); 3Department of Life Sciences, Faculty of Science and Technology, The University of the West Indies, St. Augustine 999183, Trinidad and Tobago; brian.cockburn@sta.uwi.edu; 4Laboratory Animal Medicine Unit, Directorate of Centre for Animal Health Studies, Tamil Nadu Veterinary and Animal Sciences University, Chennai 600016, Tamil Nadu, India; seenubioinfo@gmail.com; 5Department of Paraclinical Sciences, Faculty of Medical Sciences, The University of the West Indies, St. Augustine 999183, Trinidad and Tobago; chalapathi.rao@sta.uwi.edu; 6Department of Clinical Veterinary Sciences, School of Veterinary Medicine, Faculty of Medical Sciences, The University of the West Indies, St. Augustine 999183, Trinidad and Tobago; jenelle.johnson2@sta.uwi.edu

**Keywords:** intermediate frequency, electromagnetic radiation, histology, biochemical, Sprague Dawley rats

## Abstract

**Simple Summary:**

With the proliferation of electronic appliances within the past 20 years, there has been increased exposure to intermediate frequency electromagnetic radiation (IF EMR) from numerous sources. There is now considerable interest in whether these frequencies have deleterious effects on biological systems. The objective of this project was to investigate the effects of 150 kHz IF EMR on haematological, biochemical, gross, and histological profiles of vital organs of Sprague Dawley (SD) rats using standard procedures. This study concluded that whole-body exposure to 150 kHz IF EMR for two months has no adverse effects on the major organs of SD rats except the liver and lungs. In addition, an increase in serum sodium levels and a decrease in serum urea levels were observed.

**Abstract:**

Exposure to electromagnetic radiation (EMR) from intermediate frequency sources has increased exponentially in recent years. The consequences of this exposure on biological systems are prompting scientists to study the effects on human health. This current study aimed to determine the effects of intermediate frequency (150 kHz) EMR exposure on the vital organs of female Sprague Dawley rats. The EMR group (*n* = 10 animals) was exposed to a frequency of 150 kHz with an intensity of 65 ± 15 μW/cm^2^ for two months. The control group (*n* = 10 animals) was exposed to an intensity of 35 ± 15 nW/cm^2^. Haematological, histochemical, gross, and histopathological profiles of all major organs of all animals were then performed using standard procedures. All major organs generally showed no significant detectable effects in either the control or EMR groups. However, gross and histopathological examinations revealed the effects of EMR on the liver and lungs, which showed inflammatory changes without significant biochemical/haematological manifestations. In addition, a significant increase in serum sodium level and a decrease in serum urea level were also observed in the EMR group. It can be concluded that the current frequency and duration of exposure trigger the changes in the liver and lungs but are not sufficient to cause clinical and functional manifestations. Therefore, a long-term exposure study might be helpful to determine the effects of 150 kHz IF EMR on these organs.

## 1. Introduction

Electromagnetic (EM) waves from both natural and man-made sources play an essential role in regulating our daily activities. With advances in technology, people are now increasingly exposed to EM waves from various sources that emit frequencies in the intermediate frequency range (IF). IF is a form of non-ionizing radiation (NIR) that is generally considered safe because it does not contain enough energy to ionize matter but can produce heating effects. The International Commission on Non-Ionizing Radiation Protection (ICNIRP) does not provide an explicit definition for the IF range [1], while the World Health Organization (WHO) defines this range between 300 Hertz (Hz) and 10 Megahertz (MHz) [2]. The Korea Institute of Electromagnetic Engineering and Science (KIEES), which is responsible for standardizing terms related to electromagnetic fields, considers the IF range as 100 Hz to 10 MHz [3]. The same frequency range has been reported in the literature by several sources [4,5]. However, the European Commission (SCENIHR) defines the IF as between 300 and 100 kHz [6].

Devices using the IF range include computers, video terminals (VDT), tube TVs (15 kHz–25 kHz), anti-theft devices with bandwidths from several kHz to several MHz, toys with electric motors, and wireless power transmission systems (10 kHz–100 kHz, 400 kHz, and 6 MHz). This IF is becoming more common in homes and offices as it uses new, environmentally friendly technologies. Some of these new sources are induction cooktops, electronic article surveillance (EAS), and energy-saving compact fluorescent lamps [4,5,7,8,9,10].

Nowadays, many biological effects of electromagnetic waves are intensively researched. It is known that biological systems respond differently to electromagnetic waves with different frequencies and intensities [11]. With the exponential increase in the technological development of IF, human exposure has multiplied in recent years, leading to health concerns. The biological effects of this range can be divided into two main types: thermal and non-thermal. Thermal effects are induced by the absorption of energy by the skin and superficial tissues [12], while non-thermal effects depend on the amount of energy absorbed and mediate the generation of reactive oxygen species (ROS) in the mitochondria of the cell [13].

The WHO [7] provides evidence for the safety of the IF field. However, the literature has shown that EMR exposure from NIR sources can affect the biochemical processes of vital organs, leading to changes in cell membranes, an increase in peroxidation, oxidative stress, neurotransmitter synthesis and blood–brain barrier permeability, genotoxicity, carcinogenicity, and can lead to cancer with prolonged exposure [14]. There are also studies showing damage to the brain, liver, kidneys, and reproductive system [15,16,17]. In contrast, numerous epidemiological, in vivo, and in vitro studies confirm the safety of the IF [18,19,20,21]. In addition, recent innovative research has demonstrated the efficacy of this field (100 kHz–300 kHz) in disrupting mitosis in cancer cells [22] and in slowing the development of follicular cysts in polycystic ovaries (PCOs) [23,24]. However, these studies are controversial and do not lead to a clear conclusion because there are many differences in exposure duration, frequency, orientation, modulation, and power density. The magnitude of the biological effects of the IF range on vital organs is proportional to the rate of absorption in relation to the frequency of environmental exposure, which is often neglected in experimental design [25]. In our previous studies [23,24], we investigated the effect of 150 kHz EMR on rat models with PCOs and found changes in the ovaries, pituitary, and hypothalamus, as well as changes in the development of ovarian follicles and the number of cells in the anterior pituitary. Therefore, in this study, we used the same frequency to further investigate the effects on other vital organs in Sprague Dawley (SD) rats.

The aim of this study was to investigate the haematological, biochemical, gross, and histological profiles of vital organs to determine the effects of 150 kHz IF EMR on the vital organs of SD rats and so potentially aid in the development of guidelines for the safe use of IF range devices.

## 2. Materials and Methods

### 2.1. Animal Care and Husbandry

Twenty (20) healthy young adult female SD rats (12–15 weeks old) weighing 200–300 g were used for the current study. The animals were purchased from the laboratory animal facility of the School of Veterinary Medicine, The University of the West Indies, St. Augustine, Trinidad, and Tobago. The animals were housed in polypropylene cages (2 animals per cage) measuring 40 × 24 × 14 cm in a dedicated experimental room at the School of Veterinary Medicine, at an ambient temperature of 22 ± 3 °C and relative humidity of 50–60%, with a 12 h alternation of light and dark. Standard pellet food and water were provided ad libitum. All animals were acclimatized to laboratory conditions 7 days before the experiment. All animal experiments were performed in strict accordance with the Guide for the Care and Use of Laboratory Animals [26] and approved by the Campus Research Ethics Committee, The University of the West Indies, St Augustine, Trinidad, and Tobago (CREC-SA.1279/12/2021).

Rats were randomly divided into two groups (control and EMR group) created with Microsoft Excel using = RAND (), with 10 animals per group. The EMR group was continuously irradiated with 150 kHz EMR from an EMR generation system for two months (except for approximately one hour per week needed for cage changes) [23,24]. The control group was not irradiated with EMR from the device.

### 2.2. Experimental Setup and EMR-Generation System

The experimental animals were placed in a uniform electromagnetic field with a frequency of 150 kHz and an amplitude voltage of 12 V (Figure 1). The system was designed to generate EMR. The system was consistent with the previous study design at this frequency [21,23,24]. The EMR signal was generated by a Kenwood AG-203A oscillator (10 Hz–1 MHz) (Trio-Kenwood Electronic; Komagane, Japan) with the maximum possible result intensity. The field intensity within the experimental group was maintained at 0.3 V/cm^2^. The intensity of the EMR field in the cages was measured and monitored weekly using an NBM-550 broadband field meter (100 kHz–6 GHz) (Narda Safety Test Solutions GmbH, Pfullingen, Germany). The EMR cage was surrounded by aluminium foil to avoid radiation leakage from the source to the surroundings. The control cage was also surrounded by aluminium foil as an additional protective layer and placed 5 feet away from the EMR group. The intensity of the EMR field was 65 ± 15 μW/cm^2^ in the EMR group and 35 ± 15 nW/cm^2^ in the control cages. The whole room had an exposure of 0–100 nW/cm^2^. The EMR intensity was more than 1000 times higher in the exposed cages than in the control cages. The experimental setup acted as a capacitor thus storing charges within the system. The circuit was closed so it did not allow for any leakage to the surrounding objects. No other source in the room, such as lighting fixtures and electrical outlets, had EMR in the IF range used for the experiment and was not considered as interference. Cell phones were not allowed in the room because the intensity measured was around 100 nW/cm2. This meant that our intensity was 1000 times higher, but the energy absorbed by living tissue cannot be compared because of the different absorption levels. The geometry and positions of the cages, electrodes, and oscillators did not change during the experiment.

### 2.3. After 2 Months of Exposure to EMR

At the end of the exposure period, the animals were weighed and intraperitoneally sedated with ketamine hydrochloride (Dutch Farm Veterinary Pharmaceuticals, Dutch Farm International BV, Utrecht Holland) at a dose of 80 mg/kg. Once the rats were sedated, they were induced into deep anaesthesia by intraperitoneal administration of pentobarbital sodium (Kela NV, St. Lenaartseweg, Belgium) at a dose of 40 mg/kg. An amount of 3 mL of blood was collected from each animal using a standard terminal cardiac puncture protocol. Immediately after blood collection, the animals were euthanized intraperitoneally by an overdose of sodium pentobarbital at a dosage of 120 mg/kg.

### 2.4. Haematological and Biochemical Analysis

Blood was collected in Vacutainer tubes coated with ethylene diamine tetra acetic acid (EDTA) for haematological analysis. The blood parameters such as white blood cell (WBC) count, red blood cell (RBC) count, haematocrit (HCT), haemoglobin (Hb), mean corpuscular volume (MCV), mean corpuscular haemoglobin (MCH), mean cell haemoglobin concentration (MCHC), red cell distribution width (RDW), reticulocytes (REL), plasma proteins, platelets, were determined using an automated haematology analyser (ProCyte Dx™, Idexx Laboratories, Westbrook, ME, USA).

For biochemical analysis, solidified blood samples were centrifuged in non-EDTA coated tubes in a benchtop centrifuge (TJ-6, Beckman Coulter Inc., Brea, CA, USA) at 1000× *g* and room temperature for 10 min to obtain sera for analysis. The biochemical parameters of serum values of sodium (Na^+^), potassium (K^+^), sodium: potassium (Na^+^:K^+^) ratio, chloride (Cl^−^), calcium (Ca), phosphorous (P), urea, creatinine, total protein (TP), albumin, globulin, albumin: globulin (A:G) ratio, glucose and cholesterol, alkaline phosphatase (ALP), alanine aminotransferase (ALT), aspartate aminotransferase (AST), were analysed using a chemical analyser (BS 200, Mindray Medical International Company, Shenzhen, China).

### 2.5. Gross and Histopathological Analysis

Following euthanasia, a veterinarian performed a thorough necropsy examination to examine gross pathology and any reported lesions. Tissue samples of the vital organs, lung, liver, kidney, brain (hypothalamus), heart, spleen, pituitary, lymph nodes, pancreas, stomach, intestine (jejunum), skin, ovary, and uterus, were collected at necropsy examination for histopathological examination. The hypothalamus was chosen to represent the brain because it is a neuroendocrine centre that controls and regulates many body functions. This means that the hypothalamus is responsible for homeostasis and any changes in the hypothalamus can reflect on the profile of most vital organs more than any other site in the brain. The jejunum was chosen to represent the intestine because the core function of the intestine is the absorption of nutrients, which occurs mainly in the jejunum. The oestrus cycle, which affects the structure of the ovaries and uterus, was not monitored during this experiment because our previous experiments have shown that EMR exposure has no significant effect on normal cyclicity in rats. All specimens were fixed in 10% buffered neutral formalin and processed by routine histological processing. Sections were cut to 3–5 μm thickness using a rotary microtome (Finesse ME, Thermo Scientific Fisher Company, Waltham, MA, USA). Sections were stained with haematoxylin and eosin (H&E), examined, and photographed using an Olympus BX51 system microscope with CellSens Imaging Software (version 1.12) and Olympus DP71 digital camera (Olympus Corporation, Tokyo, Japan).

### 2.6. Statistical Analysis

All statistical analyses were performed using GraphPad Prism software (version 9.0) (GraphPad Software Inc, CA, USA). Data from each group were tested for equality of variance using the “F” test. If the variance of both groups was not significant, a parametric Student’s “*t*” test was performed; otherwise, a nonparametric test equivalent to Student’s “*t*” test, the Mann–Whitney test, was performed. *p* < 0.05 was considered significant. For the haematological data, the Mann–Whitney test was performed directly to compare the median value between the control and EMR groups because of the limited number of data available due to unexpected haemolysis of some samples. Biochemical parameters were expressed as mean ± SEM, whereas haematological parameters were expressed as median and mean ± SD.

## 3. Results

### 3.1. Body Weight

The 2-month whole-body exposure to 150 kHz IF EMR has no significant effect on the body weight of rats (Figure 2).

### 3.2. Haematological Analysis

Although the haematologic parameters between the control and EMR groups were not statistically significant, they suggest that EMR may have influenced leukocyte counts, as the values were almost half of those in the control group. In both the control and EMR groups, all other haematological parameters were within the reference range (Table 1).

### 3.3. Serum Biochemical Analysis

All biochemical parameters studied in the exposure groups were within the reference range for the rats. The levels of potassium, sodium: potassium, chloride, calcium, phosphorus, ALP, AST, ALT, total protein, albumin, globulin, albumin: globulin, creatinine, glucose, and cholesterol in the control and EMR groups were not significantly different (*p* > 0.05). There was a significant increase (*p* < 0.05) in the sodium level and a significant decrease (*p* < 0.01) in urea in the EMR group (Table 2).

### 3.4. Gross Pathology

The internal organs of the control and EMR groups did not show abnormal changes or lesions at necropsy except for the liver and lungs. The lungs of both groups showed mild lesions such as multifocal to diffuse congestion without exudate on the surface or cut section in most animals. The liver showed mild inflammatory changes in some animals in the EMR group (six of ten animals) (Figure 3)

### 3.5. Histopathological Analysis

Histopathologic evaluation showed normal architecture of all examined organs in the EMR group (Figure 4, Figure 5, Figure 6, Figure 7 and Figure 8), which was comparable to that of the control group. However, changes were noted in the liver and lungs. The lungs showed thickening of the alveoli with interstitial inflammatory infiltrates and mild congestion in both groups. However, the intensity was more in most of the animals in the EMR group than in the control group. The liver showed normalized liver parenchyma consisting of hepatocytes and sinusoids with central venous and portal tracts. Some of the EMR specimens showed mild inflammatory changes with lymphocytic infiltration. The kidney showed normal renal morphology. Glomeruli, tubules, and interstitium appeared normal (Figure 4). The brain showed glial cells and typical neuronal architecture. Normal cardiac muscle fibres were seen in the myocardium. Normal lymphoid follicles and sinusoids were visible in the spleen (Figure 5). The pituitary gland showed small clusters of normal-looking basophils, acidophils and chromophobes embedded between thin-walled vessels. The lymph nodes had normal architecture, consisting of normal lymphoid follicles and parafollicular tissue. The pancreas had normal pancreatic acini and ducts and pancreatic islets (Figure 6). The mucosal epithelium and glands of the stomach appeared normal. The lamina propria appeared typical. The intestine (jejunum) displayed a typical mucosa with enterocytes and goblet cells lining the villi. The epidermis and dermis of the skin appeared normal (Figure 7). The ovary displayed typical ovarian morphology, including stromal cells and regular follicular development. The endometrium of the uterus was normal with typical uterine glands and blood vessels. The smooth muscle fibres of the myometrium were normal (Figure 8).

## 4. Discussion

There is concern about the effects of increased daily exposure to EMR due to the significant role that electromagnetic radiation plays in our lives. The main goal of this study was to investigate whether 150 kHz IF EMR influences the major organs of a mammalian model when it is constantly exposed to the whole body for two months, and if so, whether these effects are associated to other health risks. This frequency was chosen because it has been proven successful in the treatment of various cancers through its antimitotic effect (Tumour Treating Fields) in preclinical settings [22]. Such reports aroused our interest in experiments with non-cancerous tissues. Since the intermediate frequency is considered a non-ionizing radiation, the authors decided to perform a short-term exposure, and if the results are significant, a long-term exposure is planned. In addition, most of the work in the literature was conducted over 6–8 weeks. Therefore, for our previous studies [23,24], an exposure of two months was chosen to study the hypothalamic–pituitary–ovarian axis, which showed no changes in the hypothalamus but observable changes in the pituitary and ovaries, but biochemical and haematological parameters were not completed in those studies. Considering these results, we wanted to further investigate the effects of this frequency on other tissues and organs for the same duration of two months.

Rats exposed to 150 kHz IF EMR for two months did not show any significant differences in body weight gain as compared to the control group of rats. However, additional parameters such as haematological, serum biochemical and gross and histopathology of the major organs were performed to determine the effects of EMR on various organs.

The haematopoietic system is the most sensitive target for studying the effects of EMR because it serves as an indicator of the overall physiological and pathological state of the body [27]. The haematological characteristics offer a higher level of predictability of damage in humans when interpreting the relevance of data from animal research [28]. In the current study, all variables were within the reference range for rats [29,30] and no statistical significance was found between the groups. Abdolmaleki et al. [31] found a decreased percentage of lymphocytes and white blood cells but an increased haemoglobin level and red blood cell count in workers exposed to EM waves, with a significant increase in mean MCV and platelet percentage. He attributed this increase to the stimulatory effect of the waves from the EMR on the division of haematopoietic stem cells, which was not observed in the present study because there were no observable significant differences in complete blood count (CBC) parameters. However, it was found that the mean value of WBC decreased in the EMR group. Considering that there is no statistical significance between the groups and the values are still within the reference range, and the limited data on haematological samples, this result needs to be confirmed by further studies.

The function of the liver is assessed by bile secretion, protein synthesis, glucose metabolism, or abnormal protein levels in the blood [32]. Some enzymes and proteins such as ALT, AST, gamma-glutamyl transferase and bilirubin have been identified as sensitive indicators of hepatocellular function [33]. Elevated serum levels of AST and ALT indicate liver damage [34], often associated with liver toxicity, hepatitis, and liver necrosis [35]. In the current study, there were no statistically significant differences between the EMR and control groups in ALT, AST, and ALP enzymes. However, histopathological examination of the livers of the EMR group showed mild lymphocytic infiltrates and haemorrhages in some animals, indicating possible liver injury or infection. This observation is consistent with reports that exposure to EMR from mobile phones/microwaves has adverse effects on the liver [36,37,38,39]. The reason for this has been suggested to be that EMR affects the cell membranes of liver hepatocytes and causes cellular damage through the action of reactive oxidative species (ROS). In addition, prolonged exposure to EMR generated by cell phones may cause selective tissue damage to the liver, and the extent of this damage is expected to increase over time [40,41,42]. Exposure to low frequencies can lead to fibrosis, steatosis, diffuse necrosis, and infiltration of a few inflammatory cells in portal areas of the liver [43].

Blood urea nitrogen and creatinine can be used to measure changes in renal function; an increase in these levels indicates impaired renal function [44]. Serum urea is significantly decreased in the current study, but creatinine, total proteins, albumin, and globulin were within the reference range. Decreased urea levels can be attributed to chronic liver insufficiency [45], but in the current study there are mild histopathological changes with normal liver biochemical values, suggesting that the decrease in serum urea levels at the frequency and duration of exposure studied is not related to liver injury. In addition, the normal creatinine levels in the present study indicate that exposure to EMR in the current study had no effect on renal function. This is confirmed by the histopathology of the kidney, which showed normal architecture without structural damage in the EMR group in the current study. However, Borzoueisileh et al. [46] reported that both BUN and creatinine levels raised non-significantly after exposure to 8 Gy of X-rays.

Sodium is important for maintaining normal cell homeostasis and regulating fluid and electrolyte balance and blood pressure. The significant increase observed in the EMR group in the present study could be due to several factors, such as alteration of dependent Na-K transmembrane ion channels, changes in cellular calcium homeostasis, increased cellular excitability, dehydration, and modulation of cellular response to stress [47,48]. However, in the current study, there were no histopathological changes in renal tissue and no change in haematological parameters. In addition, the values were still within the normal reference range for a rat, so the reason for the increase in sodium concentration is not known. The difference between the EMR group and the control group may be attributed to the biological variation of the species. Further studies are needed on the reproducibility of these results to attribute the reason for the difference to EMR exposure.

The other trace elements, such as calcium, phosphorus, potassium, and chloride, all play important and complex roles in cellular metabolism in humans and animals [49]. Despite the statistical increase in sodium levels, potassium, sodium–potassium ratio, and chloride levels were all normal, also indicating that filtration in the renal glomerulus was normal and the potassium chloride cotransporter was functioning properly [50]. In addition, the absence of renal injury was confirmed by the calcium and fasting glucose values and gross and histopathological observations in both the EMR and control groups. Calcium and phosphorus levels were found to be elevated in electrical workers who spent a large portion of their shift in substations and on power lines. These activities were associated with an increase in oxidative stress [51]. Although these emissions are harmless, it is important to remember that exposure factors such as time, wave type, and frequency can cause EMRs to be absorbed by the body through heat and trigger chemical reactions with undesirable effects [52].

Gross and histopathological examinations were used to support the haematological and biochemical findings. All organs examined showed normal gross and histologic structure, except for the lungs and liver. The liver from the EMR group showed normal architecture, comparable to that of the control group. However, some of the EMR animals (six of ten animals) showed mild inflammatory lesions with lymphocytic infiltration. In general, inflammatory lesions lead to an increase in liver enzymes as they are released from the damaged liver cells. However, in the present study, the lesions were mild in the EMR group with no significant changes in the liver enzyme levels. So, it can be suggested that the liver damage was not sufficient to lead to clinical and functional manifestations. Further, it can be concluded that two months of EMR exposure is not sufficient to cause clinicopathological manifestations, but the prolongation of this exposure may lead to clinical liver symptoms since the liver structure is already altered with current exposure levels.

Gross pathology revealed that mild lesions such as multifocal to diffuse congestion were present in the lungs of most animals in both groups. The histopathology of the lungs showed thickened alveolar walls with mild congestion in all groups, including the control groups. The presence of lesions in both groups could be due to a subclinical infection developed during the experiment. However, the intensity of the lesions was higher (maximum score of +++ in six animals) in the EMR group than in the control group (maximum score of ++ in 3 animals) which indicated that EMR exposure has effects on inflammation. The previous studies on the IF EMR up to 7.5 kHz to 150 kHz with different intensities have found no effect on the lung [53]. Radiofrequency exposure showed effects on the lung such as thickening of the alveolar wall and fibrosis and vascular congestion and lymphatic infiltration and inflammatory changes [54,55]. In the present study, the effect of EMR on the lungs need to be confirmed by repeating the work.

The other organs examined did not show major gross/histopathological lesions in either group in the current study. In contrast, many reports found changes in the organs due to EMR exposure, such as gross distortion of the architecture of the heart muscle, gradual loss, and degeneration of epithelial cells in the kidney at RF exposure [37]. Damage to the brain, liver, and kidney of rats from EMR exposure was also confirmed by Sharma et al. [14]. Damage to the spleen of mice [56], increased basophil reactivity, and increased secondary follicle formation leading to atresia [23] have also been observed in the literature.

Most studies have reported that IF EMR has no observable adverse effects [53]. Although some studies have reported various biological effects, these studies were not independently replicated and were not dependent on the level of IF EMR exposure, which is a major limitation. In addition, the animal studies did not identify an exposure threshold for IF EMR. It is crucial to take into consideration the exposure time, wavefront, frequency, field intensity, and specific absorption rate to better comprehend the effects of exposure to 150 kHz EMR. We suggest that future experimental studies continue to monitor animal health effects associated with IF EMR to provide adequate and sufficient information for a meaningful safety assessment.

## 5. Conclusions

This study concluded that two months of whole-body exposure to 150 kHz IF EMR had no adverse effects on the major organs of Sprague Dawley rats, except for the liver and lungs, which showed inflammatory changes without biochemical/haematological manifestations. In addition, an increase in serum sodium levels and a decrease in serum urea levels were noted. Thus, the current frequency and duration of exposure trigger changes in the liver and lungs but are not sufficient to cause clinical and functional manifestations. Therefore, a long-term exposure study might be helpful to determine the effects of 150 kHz IF EMR on these organs.

## Figures and Tables

**Figure 1 biology-12-00310-f001:**
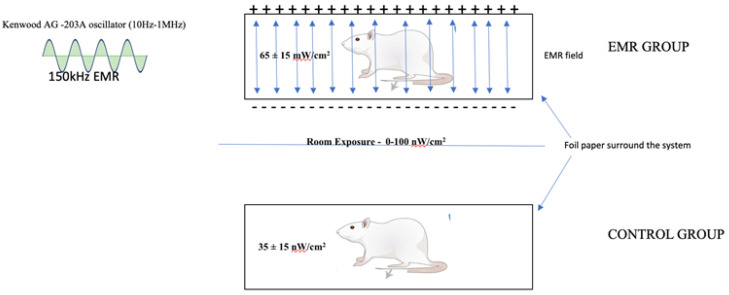
Schematic diagram of experiment setup and EMR generation system.

**Figure 2 biology-12-00310-f002:**
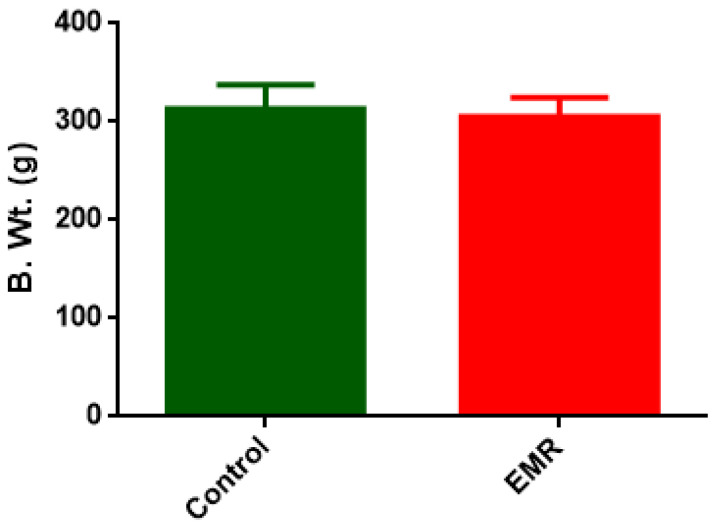
Effect of 150 kHz IF EMR on mean body weight of SD rats after 2-month whole-body exposure.

**Figure 3 biology-12-00310-f003:**
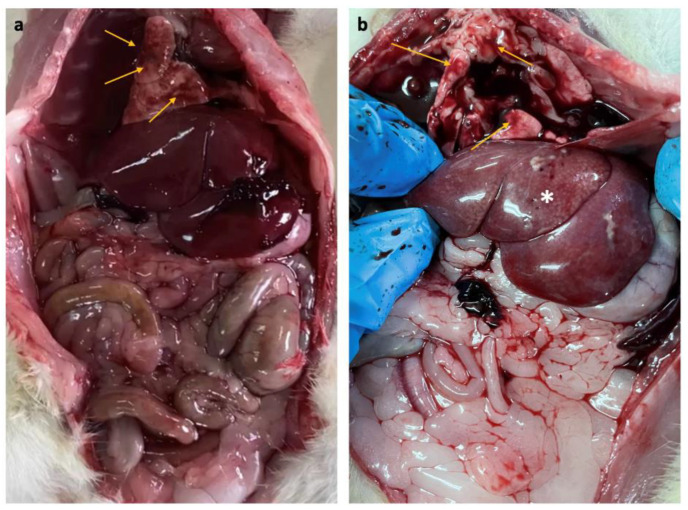
(**a**) Gross pathology of rats after two months of whole-body exposure to 150 kHz IF EMR showing multifocal and diffuse congestion (arrows) in the lung of the control and EMR groups, (**b**) multifocal and diffuse congestion (arrows) in the lung, inflammatory changes in the liver (asterisk) were observed only in the EMR group.

**Figure 4 biology-12-00310-f004:**
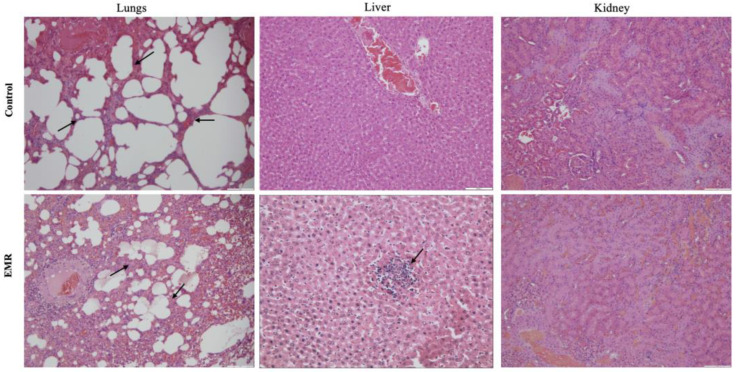
The effect of 150 kHz IF EMR on the histopathology of lung, liver, and kidney of rats in the control and EMR groups after 2 months of whole-body exposure (H&E × 200). The lungs displayed congestion and thickened alveolar walls (arrows) in both groups, but the intensity was more in EMR group. The liver showed typical hepatic cords with sinusoids and the central vein in both groups but mild parenchymal lymphatic infiltration (arrow) was found in the EMR group (H&E × 400). The kidney showed normal renal corpuscles, tubules, and collecting ducts in both groups.

**Figure 5 biology-12-00310-f005:**
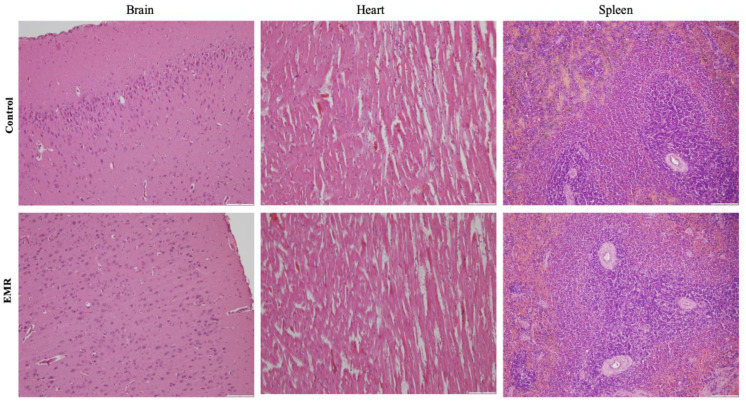
The effect of 150 kHz IF EMR on the histopathology of brain, heart, and spleen of rats in the control and EMR groups after 2 months of whole-body exposure. In the brain, glial cells and typical neuronal architecture were visible. In the myocardium, normal myocardial fibres could be detected. In the spleen, there were sinusoids and typical lymphoid follicles. (H&E × 200).

**Figure 6 biology-12-00310-f006:**
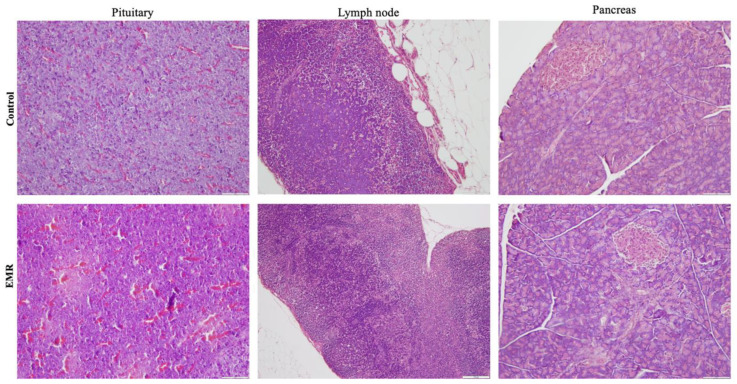
The effect of 150 kHz IF EMR on the histopathology of the pituitary, lymph node, and pancreas of rats in the control and EMR groups after 2 months of whole-body exposure. In the pituitary gland, small groups of normal-looking secretory cells were visible between thin-walled blood vessels. The lymph nodes showed typical lymphoid follicles and parafollicular tissue and were structurally normal. The pancreas showed typical pancreatic islets, ducts, and acini. (H&E × 200).

**Figure 7 biology-12-00310-f007:**
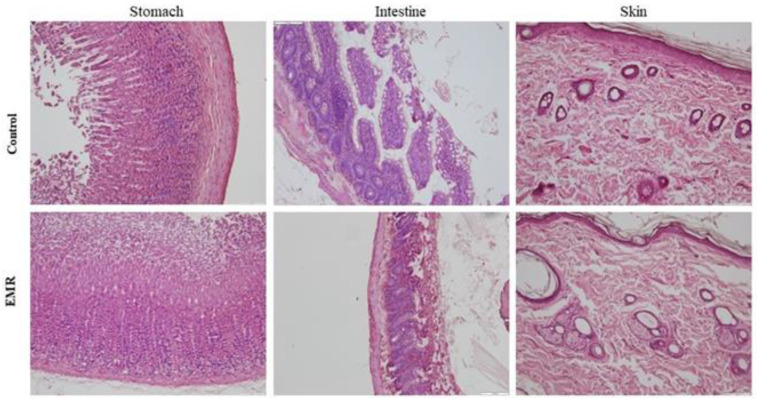
The effect of 150 kHz IF EMR on the histopathology of the stomach, intestine (jejunum), and skin of rats in the control and EMR groups after 2 months of whole-body exposure. The mucosal epithelium and glands of the stomach appeared normal. The lamina propria appeared to be healthy. The intestine showed a typical mucosa with enterocytes and goblet cells lining the villi. The epidermis and dermis of the skin appeared normal with typical glands (H&E × 200).

**Figure 8 biology-12-00310-f008:**
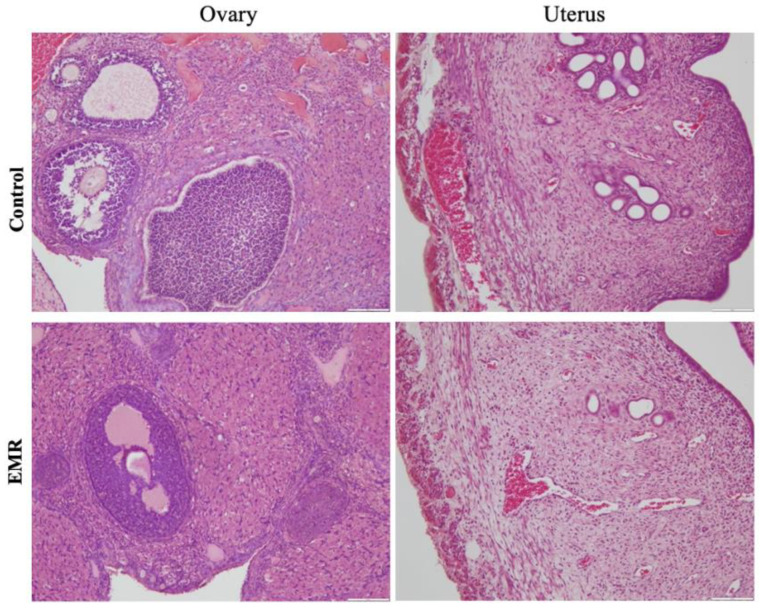
The effect of 150 kHz IF EMR on the histopathology of the ovary and uterus and pancreas of rats in the control and EMR groups after 2 months of whole-body exposure. The ovary showed typical ovarian morphology, including stromal cells and regular follicle formation. The uterine endometrium was normal, and myometrial smooth muscle fibres, uterine glands, and blood vessels were all regular. (H&E × 200).

**Table 1 biology-12-00310-t001:** Effect of 150 kHz IF EMR on haematological parameters in SD rats after 2-month whole body exposure.

Parameters	Control (Median)	EMR (Median)	Mann-Whitney *p*-Value	Statistical Significance	Control (Mean ± Standard Deviation)	EMR (Mean ± Standard Deviation)
WBC (×10^9^/L)	8.6	4.3	0.1	NS	8.1 ± 1.75	4.4 ± 1.33
RBC (×10^12^/L)	7.5	7.9	0.2	NS	7.5 ± 0.0	7.9 ± 0.01
HCT (L/L)	0.43	0.39	0.2	NS	0.42 ± 0.01	0.39 ± 0.01
Hgb (g/L)	134.0	139.5	0.4	NS	134.0 ± 5.00	139.5 ± 2.12
MCV (fL)	51.1	49.0	0.2	NS	52.0 ± 1.70	49.0 ± 1.41
MCH (pg)	18.0	17.6	0.2	NS	18.5 ± 1.01	17.6 ± 0.28
MCHC (g/L)	352.0	356.5	0.8	NS	357.3 ± 10.12	356.5 ± 4.95
RDW (%)	20.5	21.0	0.5	NS	20.5 ± 0.55	21.0 ± 0.71
Retic (%)	2.4	2.2	0.4	NS	2.6 ± 0.46	2.2 ± 0.21
Plasma protein (g/L)	81.0	76.0	0.2	NS	80.3 ± 2.08	76.0 ± 0.00
Platelets (×10^9^/L)	647.0	714.0	0.4	NS	628.3 ± 94.39	714.0 ± 74.95

NS—Not Significant; *n* = 3.

**Table 2 biology-12-00310-t002:** Effect of 150 kHz IF EMR on biochemical parameters in SD rats after 2-month whole body exposure.

Parameters	Control (Mean ± SEM)	EMR (Mean ± SEM)	*p*-Value	Statistical Significance
Serum Na^+^ (mmol/L)	138.5 ± 0.9	142.2 ± 1.0	0.03	*
Serum K^+^ (mmol/L)	7.2 ± 1.2	6.5 ± 0.6	0.60	NS
Serum Na^+^: K^+^ ratio	20.7 ± 3.0	23.0 ± 2.7	0.59	NS
Serum Cl^−^ (mmol/L)	101.4 ± 1.9	101.7 ± 0.2	0.88	NS
Serum Ca^+^ (mmol/L)	2.62 ± 0.06	2.50 ± 0.05	0.17	NS
Serum phosphorus (mmol/L)	2.05 ± 0.20	2.11 ± 0.12	0.79	NS
Urea (mmol/L)	7.7 ± 0.7	5.8 ± 0.2	0.01	**
Creatinine (µmol/L)	55.1 ± 7.3	55.9 ± 3.0	0.91	NS
Total protein (g/dL)	69.4 ± 2.3	68.9 ± 1.0	0.82	NS
Albumin (g/dL)	32.0 ± 0.7	31.3 ± 0.5	0.46	NS
Globulin (g/dL)	38.7 ± 2.4	37.4 ± 0.8	0.57	NS
Albumin: Globulin ratio	0.76 ± 0.09	0.84 ± 0.02	0.33	NS
Glucose (mmol/L)	12.8 ± 2.7	10.6 ± 1.0	0.36	NS
Cholesterol (mmol/L)	1.7 ± 0.1	1.6 ± 0.1	0.27	NS
ALP (U/L)	189.8 ± 11.9	161.0 ± 10.2	0.10	NS
ALT (U/L)	547.4 ± 285.8	251.8 ± 118.0	0.28	NS
AST (U/L)	604.0 ± 252.1	397.1 ± 156.1	0.47	NS

NS—Not Significant; * *p* < 0.05; ** *p* < 0.01; *n* = 10.

## Data Availability

All data generated and analysed during this study are included in this published article.

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
