# Peer review of "Effects of Intermediate Frequency (150 kHz) Electromagnetic Radiation on the Vital Organs of Female Sprague Dawley Rats"

_biology, 2023, doi:10.3390/biology12020310_

Round 1

Reviewer 1 Report

In this manuscript the effects of a two-month exposure to intermediate frequency electromagnetic radiation (IM-EMR) are presented in a descriptive manner, showing the anatomical and biochemical changes in the rat organism. The observed changes are useful because they may indicate the sensitivity and response of organs to IM-EMF i.e. exposure to IM-EMR could potentially threaten homeostasis or affect the health of the organism in the long-term and is an area of intensive study. However, several topics need to be explained and improved.   

 Introduction

In Introduction, which is concise and informative, in first sentence what is the differences between human and man-made sources?

 Materials and Methods

Line 121 is missing a reference.

In Experimental setup and EMR-generation system - some kind graphics presentation of EMR-generation system should be attached with clearly presented field intensity (or magnetic flux density) and changes of EMP intensity depending on the radiation source (distance).

 Results

In Hematological Analysis in table 1 WBC count showed absence of statistically significant differences (p<0,1). Despite the fact that the mean value is reduced, the lack of statistical significance cannot lead to the conclusion that exposure to the magnetic field causes a decrease in WBC. This result cannot be presented in the abstract and discussed.

In Figure 3 on the presented magnification, I was not able to see mild parenchymal infiltration in the liver.

 Discussion

According to the previous note (about the WBC result), the discussion of mild histological changes in the liver and lungs, as well as biochemical parameters (Na and urea concentration) should additionally emphasize the possible consequences of the described changes, and thus the meaning and significance of the entire study.

Author Response

Below are the reviewers’ responses to the comments. We have revised the entire paper to better express on behalf of the authors, I would like to express my sincere thanks for the elite review. If there are any further changes, please feel free to contact us.

Reviewer 1 comments

  • In Introduction, which is concise and informative, in first sentence what is the differences between human and man-made sources?

This was an error. It was changed to natural and man-made sources.

  • Line 121 is missing a reference. In Experimental setup and EMR-generation system - some kind graphics presentation of EMR-generation system should be attached with clearly presented field intensity (or magnetic flux density) and changes of EMP intensity depending on the radiation source (distance).

The missing reference was added. A graphical representation has been added to represent the experimental set-up as described in the procedure.

  • Results: In Hematological Analysis in table 1 WBC count showed absence of statistically significant differences (p<0,1). Despite the fact that the mean value is reduced, the lack of statistical significance cannot lead to the conclusion that exposure to the magnetic field causes a decrease in WBC. This result cannot be presented in the abstract and discussed.

Agreed with the comments of the reviewer, and changes were made in the abstract and discussion.

  • Discussion: According to the previous note (about the WBC result), the discussion of mild histological changes in the liver and lungs, as well as biochemical parameters (Na and urea concentration) should additionally emphasize the possible consequences of the described changes, and thus the meaning and significance of the entire study.

The necessary corrections were made in the discussion and accordingly the conclusion and summary were altered.

Reviewer 2 Report

I have carefully read the manuscript by Sundaram et al, which aim was to evaluate body effects of EMR exposure in a rat model. The study is interesting since it explored a common and current scientific question that can have important consequences for human medicine. Moreover, even if the study is relatively simple in its design, it is well conducted and it seems to well investigate its scientific purpose. However the manuscript in its current form needs some important revisions before to be considered suitable for the publication in the Journal.

First, I found that the discussion did not face the main findings of this study in a deep and correct way. Here below some examples:

-          concerning urea levels, the authors should better explain (or try to explain) why if all kidney parameters did not change between groups, urea was decreased in EMR group.

-          Even more important: concerning sodium level change, the authors concluded that this changes could not be caused by EMR exposure since also in EMR group the levels remained in the biological range. In my opinion, this explanation is not satisfactory. Even if the EMR group presented levels into the range, a statistically significant increase has been observed, which suggests that a biological phenomenon linked to EMR exposure happened during the two months. This deserves a deeper explanation.

-          Concerning liver histological lesions, the authors stated that the mild lesions could be the reason of the low hepatic enzymes levels. This is not coherent, since liver enzyme should be increased by the presence of lesions, as consequence of its liberation from damaged cells. The authors also should explain that the liver was mildly affected by EMR exposure, but that these cellular damages were not enough developed to lead to a clinical and functional manifestation. This point is important because it opens to other perspectives: two month of EMR exposure are not enough to can observe clinical liver pathology manifestation, but since the liver is already altered, the prolongation of this exposure could result in clinical liver symptoms. This perspectives partially alters yours conclusions, because it could suggest that prolonged EMR exposure can effectively produce differences among the two groups.

-          Concerning lungs, the authors stated that these lesions were not induced by EMR exposure since they were present in both groups, which sounds logical. However, an explanation is necessary. Moreover, the photos about these lesions clearly showed a different intensity between the control and the EMR group, with this latter presenting a more important increase of the thickness of the alveolar walls. To these images, I’m not sure that there is not a difference between group about lung lesion intensity. Please clarify or provide other more representative photos.

Other minor flaws are reported here below:

-          The English should be carefully and thoroughly reviser by a native speaker. In particular the two summaries.

-          Can you better explain why did you chose this frequency and two months as exposure period? Even if it is the same design of your previous studies, this is a new article and readers should easily find this information.

-          A similar point; why did you chose to analyze only  hypothalamus for brain and jejunum for the guts?

-          Table 1: please provide also mean values and SD, it could help to understand why there is not statistical difference between controls and EMR rats for WBC.

Author Response

The authors thanks the reviewers for their critical comments on the manuscript. Below gives the details of the edits made to the manuscript.

Reviewer 2 comments

  • First, I found that the discussion did not face the main findings of this study in a deep and correct way. Here below some examples:  concerning urea levels, the authors should better explain (or try to explain) why if all kidney parameters did not change between groups, urea was decreased in EMR group.

The authors tried to explain the decrease in serum urea in the discussion with the light of the results as Decreased urea levels can be attributed to chronic liver insufficiency [45], but in the current study there are mild histopathological changes with normal liver biochemical values, suggesting that the decrease in serum urea levels at the frequency and duration of exposure studied is not related to liver injury. In addition, the normal creatinine levels in the present study indicate that exposure to EMR in the current study had no effect on renal function. This is confirmed by the histopathology of the kidney, which showed normal architecture without structural damage in the EMR group in the current study.

  • Even more important: concerning sodium level change, the authors concluded that this changes could not be caused by EMR exposure since also in EMR group the levels remained in the biological range. In my opinion, this explanation is not satisfactory. Even if the EMR group presented levels into the range, a statistically significant increase has been observed, which suggests that a biological phenomenon linked to EMR exposure happened during the two months. This deserves a deeper explanation.

This has been modified in the discussion as follows, “However, in the current study, there were no histopathological changes in renal tissue and no change in hematological parameters. In addition, the values were still within the normal reference range for a rat, so the reason for the increase in sodium concentration is not known. The difference between the EMR group and the control group may be at-tributed to the biological variation of the species. Further studies are needed on the re-producibility of these results to attribute the reason for the difference to EMR exposure.

  • Concerning liver histological lesions, the authors stated that the mild lesions could be the reason of the low hepatic enzymes levels. This is not coherent, since liver enzyme should be increased by the presence of lesions, as consequence of its liberation from damaged cells. The authors also should explain that the liver was mildly affected by EMR exposure, but that these cellular damages were not enough developed to lead to a clinical and functional manifestation. This point is important because it opens to other perspectives: two month of EMR exposure are not enough to can observe clinical liver pathology manifestation, but since the liver is already altered, the prolongation of this exposure could result in clinical liver symptoms. This perspectives partially alters yours conclusions, because it could suggest that prolonged EMR exposure can effectively produce differences among the two groups.

Agreed with the reviewer’s suggestion. The necessary statements were included in the discussion as per the comment.

  • Concerning lungs, the authors stated that these lesions were not induced by EMR exposure since they were present in both groups, which sounds logical. However, an explanation is necessary. Moreover, the photos about these lesions clearly showed a different intensity between the control and the EMR group, with this latter presenting a more important increase of the thickness of the alveolar walls. To these images, I’m not sure that there is not a difference between group about lung lesion intensity. Please clarify or provide other more representative photos.

The histopathology of the lungs showed thickened alveolar walls with mild congestion in all groups, including the control groups. The presence of lesions in both the groups could be due to prevailing subclinical infection during experiment. However, the intensity of the lesions was higher (maximum score of +++ in six animals) in the EMR group than the control group (Maximum score of ++ in 3 animals) which indicated that EMR exposure have its effects in the inflammation. Therefore, the discussion/conclusion was changed accordingly.

  • Other minor flaws are reported here below: The English should be carefully and thoroughly reviser by a native speaker. In particular the two summaries.

The manuscript has been revised for English.

  • Can you better explain why did you chose this frequency and two months as exposure period? Even if it is the same design of your previous studies, this is a new article and readers should easily find this information.

The following paragraph was included at the beginning of the discussion.

This frequency was chosen because it has been proven successful in the treatment of various cancers through its antimitotic effect (Tumor Treating Fields) in preclinical settings [22]. Such reports aroused our interest in experiments with non-cancerous tissues. Since the intermediate frequency is considered a non-ionizing radiation, the authors decided to perform a short-term exposure, and if the results are significant, a long-term exposure is planned. In addition, most of the work in the literature was conducted over 6-8 weeks. Therefore, for our previous studies [23, 24], an exposure of two months was chosen to study the hypothalamic-pituitary-ovarian axis, which showed no changes in the hypothalamus but observable changes in the pituitary and ovaries. In light of these results, we wanted to further investigate the effects of this frequency on other tissues and organs for the same duration of two months.

  • A similar point; why did you chose to analyze only  hypothalamus for brain and jejunum for the guts?

The hypothalamus was specifically chosen because it is a neuroendocrine center that controls and regulates many body functions. This means that the hypothalamus is responsible for homeostasis and any changes in the hypothalamus can reflect on the profile of most vital organs than any other site in the brain. The jejunum was chosen to represent the intestine because the core function of absorption of nutrients is through the villi. It is included in the materials and methods

Table 1: please provide also mean values and SD, it could help to understand why there is not statistical difference between controls and EMR rats for WBC

The values were added in the table 1 as suggested.

Reviewer 3 Report

This study presents interesting data on the effects of electromagnetic radiation from intermediate frequencies (150 kHz) in female rat models.

A few comments/questions:

1.  Was the estrous cycle of the female rats considered in the analysis?

2.  An image or drawing of the experimental setup would be helpful in understanding the experiments. How did the authors ensure a uniform spatial distribution of the EMR in the cage?

3. What is the susceptibility of the food placed in cages to the EMR radiation?  Could these cause secondary effects on the animal physiology?

4. There are several spelling/grammar mistakes and awkward sentences (i.e ln 18) in the simple summary and abstract that need to be addressed.

5. The lack of statistical significance for WBCs is intriguing. Can the authors show the distribution and variance in this cohort-population? Have the authors characterized the WBC populations i.e Wright's staining?

6. Why was the parameter 65 uW/cm2 chosen? How does this compare to i.e cell phone, microwave and powerline radiation?

Author Response

Reviewer 3 comments

  • Was the estrous cycle of the female rats considered in the analysis?

No. The estrus cycle of rats was not considered in this experiment because EMR did not show significant changes in the estrus cycle in our previous studies. It is now stated in the materials and methods.

  • An image or drawing of the experimental setup would be helpful in understanding the experiments. How did the authors ensure a uniform spatial distribution of the EMR in the cage?

A schematic diagram (Figure 1) was added to the manuscript to show the field distribution. An electric field was generated by producing 150 kHz. The generated field acted like a capacitor and stored the energy generated by the source (represented by the field lines). The field would be uniform throughout the interior of the cages exposed to the EMR. These charges could not escape the system unless a connection was made to other devices or objects in the room. The circuit was closed so that the flow of charges within the experimental cages was maintained. This description has been added to the method.

  • What is the susceptibility of the food placed in cages to the EMR radiation?  Could these cause secondary effects on the animal physiology?

This point is a good suggestion that can be taken up in further studies by exposing only the diet and administering the exposed diet to the rats and studying their effects. This study and our previous studies showed no effects on body weight. Also, when we studied the hypothalamus and particularly the neurons around the 3rd ventricle, there were no changes in these neurons that are responsible for energy balance and metabolism.

  • There are several spelling/grammar mistakes and awkward sentences (i.e ln 18) in the simple summary and abstract that need to be addressed.

The entire manuscript checked for grammer/spelling and revised.

  • The lack of statistical significance for WBCs is intriguing. Can the authors show the distribution and variance in this cohort-population? Have the authors characterized the WBC populations i.e Wright's staining.

The mean and standard deviation of the WBC is given below and added in Table 1 according to another reviewer's comment. Because limited data (n=3) are available for these parameters, we need to examine these parameters in future studies to determine the reproducibility of the observed changes. The authors did not characterize the WBC populations by Wright's staining. We will do so in our future work on the same frequency with long-term exposure when we encounter these results.

Control

EMR

Mean 

SD

Mean 

SD

WBC X 109 /L

8.1

1.75

4.4

1.33

  • Why was the parameter 65 uW/cm2 chosen? How does this compare to i.e cell phone, microwave and powerline radiation?

This power density parameter was not chosen. A The frequency of 150 kHz was chosen because it is widely used in the treatment of various cancer cell lines (tumor treating fields). Measurement of the power intensity of the field in which we placed the animals yielded this value. Simultaneous monitoring of the room yielded a separate value. When we calculated this value, as indicated in the methods, we found that the value was 1000 times lower than when the animals were placed in the field. With this concept, we were able to set up a control room in the same room at a distance of 5 feet, and aluminum was added as a safety precaution to protect against any leaks. The power intensity of cell phones is around 100 nW when idle and 300 nW during a call. So our intensity is 1000 times higher, but 150 kHz absorbs much less compared to 1 MHz cell phone radiation. We have much higher intensity at 150 kHz, but the energy absorbed by living tissue cannot be compared because of the different absorption levels. This was added to the manuscript.

Round 2

Reviewer 2 Report

I thank the authors for having modified the text according to reviewers' suggestion. In my opinion, the manuscript is now suitable for the publication in the journal Biology. 

Reviewer 3 Report

The manuscript is much improved and is recommended for publication.